# Axl Expression in Renal Mesangial Cells Is Regulated by Sp1, Ap1, MZF1, and Ep300, and the IL-6/miR-34a Pathway

**DOI:** 10.3390/cells11121869

**Published:** 2022-06-09

**Authors:** David E. Adams, Yuxuan Zhen, Xiaoyang Qi, Wen-Hai Shao

**Affiliations:** 1Division of Rheumatology, Allergy, and Immunology, Department of Internal Medicine, College of Medicine, University of Cincinnati, Cincinnati, OH 45267, USA; adamsdv@ucmail.uc.edu (D.E.A.); zhenyn@ucmail.uc.edu (Y.Z.); 2Division of Hematology/Oncology, Department of Internal Medicine, College of Medicine, University of Cincinnati, Cincinnati, OH 45267, USA; xiaoyang.qi@uc.edu

**Keywords:** Axl, mesangial cell, transcription factor, expression regulation, Sp1, Ap1, miRNA-34a, G-quadruplex

## Abstract

Axl receptor tyrosine kinase expression in the kidney contributes to a variety of inflammatory renal disease by promoting glomerular proliferation. Axl expression in the kidney is negligible in healthy individuals but upregulated under inflammatory conditions. Little is known about Axl transcriptional regulation. We analyzed the 4.4 kb mouse *Axl* promoter region and found that many transcription factor (TF)-binding sites and regulatory elements are located within a 600 bp fragment proximal to the translation start site. Among four TFs (Sp1, Ap1, MZF1, and Ep300) identified, Sp1 was the most potent TF that promotes Axl expression. Luciferase assays confirmed the siRNA results and revealed additional mechanisms that regulate Axl expression, including sequences encoding a 5′-UTR mini-intron and potential G-quadruplex forming regions. Deletion of the *Axl* 5′-UTR mini-intron resulted in a 3.2-fold increases in luciferase activity over the full-length UTR (4.4 kb *Axl* construct). The addition of TMPyP4, a G-quadruplex stabilizer, resulted in a significantly decreased luciferase activity. Further analysis of the mouse *Axl* 3′-UTR revealed a miRNA-34a binding site, which inversely regulates Axl expression. The inhibitory role of miRNA-34a in Axl expression was demonstrated in mesangial cells using miRNA-34a mimicry and in primary kidney cells with IL-6 stimulated STAT3 activation. Taken together, Axl expression in mouse kidney is synergistically regulated by multiple factors, including TFs and secondary structures, such as mini-intron and G-quadruplex. A unique IL6/STAT3/miRNA-34a pathway was revealed to be critical in inflammatory renal Axl expression.

## 1. Introduction

Axl, together with Tyro-3 and Mer, constitutes the TAM family of receptor tyrosine kinases that share a common ligand, the growth arrest specific protein 6 (Gas6) [1]. Receptors from this family contribute to apoptotic clearance, immune homeostasis, and survival/proliferation [2]. TAM receptors exhibit broad overlapping and unique expression patterns in tissues [3]. Tyro-3 is mostly expressed in the nervous system; Mer is expressed in the hematopoietic lineages, including monocytes/macrophages, platelets, NK cells, et al.; Axl is expressed ubiquitously at a low level, with notable levels found in monocytes/macrophages, platelets, along with other organs. The importance of Gas6/Axl pathway in renal inflammation has been demonstrated by several animal models of glomerulonephritis with genetic disruption of Gas6 and Axl, and small molecule inhibitor targeted therapy [4,5,6]. However, Axl expression in the kidney is only upregulated under inflammatory conditions. The mechanisms regulating Axl expression in the kidney have not been fully understood.

Axl is important in cancer survival, and its expression has been extensively studied in malignant cells [7]. The human Axl promoter region is GC rich, and a minimal ~600 bp region is sufficient for basal *Axl* promoter activity. Five Sp-binding sites in this region constitutively drive cancer cell Axl expression [8]. Similarly, the mouse *Axl* gene promoter region is also GC-rich. We previously showed that treatment of mesangial cells with mithramycin (binds to GC-rich regions) diminished Axl expression [5]. GC-rich mouse *Axl* promoter also harbors multiple transcription factor (TF) binding sites. SiRNA-targeted Sp1 in mouse mesangial cells resulted in a robust reduction in Axl expression [5]. Recently, a variety of mechanistic studies have pointed to DNA and RNA sequences with tandem G-rich repeats that can fold into a tetraplex structure called G-quadruplex, which consists of a stack of four guanine residues interconnected by Hoogsteen base pairs [9]. Such structures may have both positive and negative effects on gene expression.

In this study, we characterized the mouse *Axl* promoter and identified binding sites for various TFs. Deletion of these sites had profound effects on *Axl* gene expression. We further showed that deletion of the mini-intron and stabilization of the potential G-quadruplex forming regions in the ~600 bp proximal region greatly affected *Axl* promoter activity. The results presented here are the first to show that rodent *Axl* expression is modulated by several secondary structures in addition to the similarly identified TFs in human cancer cells. The IL6/STAT3 pathway inhibited miRNA-34a was first identified to be responsible for Axl expression under inflammatory conditions.

## 2. Materials and Methods

### 2.1. Reagents

Synthetic miRNA mimics and siRNAs were obtained from Life Technologies (Carlsbad, CA, USA). All restriction enzymes were purchased from New England Biolabs (Ipswich, MA, USA). Anti-Axl Ab (1:2000) was from R&D biosystems (Minneapolis, MN, USA). Anti-STAT3 (1:2000), anti-pSTAT3 (1:1000), and anti-GAPDH (1:1000) Abs were from Cell Signaling Technology (Danvers, MA, USA). Anti-actin (1:1,000,000) Ab was from Abcam (Cambridge, MA, USA).

### 2.2. Analysis of the Axl Promoter and 5′-UTR

A 4.4 kb region encompassing the *Axl* 5′-UTR was downloaded from NCBI Genbank and cross analyzed with the PROMO (version 3) and pDRAW32 (version 1.1.146), online freeware programs [10,11].

### 2.3. SiRNA Mediated TF Knockdown and Western Blot Analysis

Mesangial cells (ATCC, established from normal 4-week-old mice (C57BL/6J × SJL/J), within 5 passages) [12]) were sub-cultured in a 6-well plate (2 × 10^5^ cells per well) the day before transfection. Thirty pM of siRNA were mixed with 9 μL of lipofectamine RNAiMAX (Life Technologies, Carlsbad, CA, USA) in Opti-MEM Medium and added to each well of the culture. Cells were incubated with the siRNA mixture for 2 days at 37 °C. Cells were then washed twice with cold PBS and lysed in RIPA buffer. Cell lysates were separated onto SDS-PAGE gel and Western blotted for the target proteins. All siRNAs were purchased from Thermo Fisher (Waltham, MA, USA). The catalog numbers are as follows: siSTAT3 (4392420-s74452), siAxl (4390771-S76975), siSp1 (4390771-s74197), siGAPDH (4390849), siEP300 (4390771-s116225), siMZF (4390771-s99637), siAP1 (4390771-s201552), siC-Rel (4390771-s201924), and miR-34a mimic (4464066-mc19474).

### 2.4. Generation of Luciferase Reporter Constructs

Genomic DNA was isolated from C57BL/6 mice tail digestion using QIAGEN’s DNeasy Blood & Tissue kit (Germantown, MD, USA). The 4.4 kb *Axl* 5′-UTR PCR fragment was cloned into the luciferase vector, pGL4.0-UBCmin-luc2 (KA1390, Addgene, Watertown, MA, USA) using the EcoRI and XhoI restriction sites. To ensure high PCR fidelity, Takara’s Prime STAR GXL DNA polymerase (Takara Bio, San Jose, CA, USA) was used to amplify the *Axl* promoter sequences. Three Primers were designed for this purpose. Primer L1: 5′-TTCCATCTCGAGCATGCTTTCCCCTCATGT-3′, Primer R1: 5′-TACGTGTCGAATTCCTGGGATTTGGCACTG-3′ and Primer R2: 5′-TGGGTCAGAATTCGAAGTTGATGGGCACCT-3′. The pair L1-R1 generated a 4334 bp PCR product that lacks the *Axl* 5′ UTR mini-intron region (designated Axl-Δintron) and the L1-R2 primer pair generated the 4389 bp PCR product (4.4 kb full length, Axl-UTR) that includes the mini-intron region. The full length *Axl* 5′-UTR sequence was verified by full plasmid DNA sequencing (Appendix A) (CD Genomics, New York, NY, USA). The cloned *Axl* 5′-UTR region contains 30 AG repeats, but published data indicate 32–33 AG repeats at the same region (GenBank JN964011.1 and AY278439.1). A series of different sizes of the *Axl* promoter region were created by restriction digestion of the full-length 4.4 kb fragment using different restriction enzymes. The cut DNA ends were then blunted with either DNA polymerase I Klenow fragment or S1 nuclease. Re-ligation of the gapped DNAs was performed at 16 °C with T4 ligase. The size and integrity of each construct were confirmed by agarose gel electrophoresis.

### 2.5. Luciferase Assays in Hela Cells

Hela cells from ATCC (within 10 passages), cultured in DMEM medium containing 10% FBS, and 2 mM L-glutamine (Gibco, Life Technologies, Carlsbad, CA, USA), were seeded into 24-well plates at 7.5 × 10^5^ cells per well. The next day, 128 ng of the full-length pGL4-Axl plasmid was diluted into Opti-MEM medium containing Lipofectamine 3000 plus P3000 reagent (Invitrogen, Waltham, MA, USA), transfected alone or co-transfected with 125 ng of transcription factor vector (Ex-M94, GeneCopoeia, Rockville, MD, USA). Twenty-four hours after transfection, cells were lysed with equal volumes (0.5 mL) of Steady-Glo luciferase reagent (Promega, Madison, WI, USA). The results were read within 60 min using an EnVision multimode plate reader (PerkinElmer, Waltham, MA, USA). The expression data were analyzed with GraphPad Prism 9.2 (San Diego, CA, USA).

### 2.6. MicroRNA-34a (miR-34a) Analysis in the Kidney of Nephritic Mice and in Cell Cultures

Nephritis was induced in WT C57BL/6 mice with anti-glomerular basement membrane (GBM) as previously described [6,13]. For microRNA analysis, total RNA from the kidney cortex samples were prepared with the miRNeasy Mini Prep kit (Qiagen, Hilden, North Rhine-Westphalia, Germany). Mature miR-34a levels were then analyzed by RT-PCR with the TaqMan miRNA-assay and pre-made primers (482747_mir) from Life Technologies. Results were normalized using the 2^−ΔΔCt^ method relative to the mouse kidney specific miRNA normalizer (481757_mir) recommended by Applied Biosystems (Waltham, MA, USA). The C_T_ of each test message was first normalized using the C_T_ for 481757_mir, assayed in the same sample. Fold changes were then calculated using the relative C_T_ method: fold change = 2^(normalized C^_T_
^from experimental mice − normalized C^_T_
^from control mice)^.

For the miR-34a mimic experiment, mesangial cells were seeded into 6-well plate (5 × 10^5^ cells per well) at day 0. Six to 12 ng of miR-34a mimics (Life Technologies, Carlsbad, CA, USA) were mixed with Opti-MEM and added into the lipofectamine/MEM. The mixture was then incubated at room temperature for 5 min and added into mesangial cell cultures at day 1. Axl expression was analyzed by Western blotting at day 3.

### 2.7. IL-6 Treatment

Primary kidney cells (Cell Biologics, Chicago, IL, USA) were cultured in 6-well plates at a density of 1 × 10^6^ cells per well. Recombinant mouse IL-6 (rmIL-6, R&D Biosystems, Minneapolis, MN, USA) was added into the cultures at 0, 10, 20, and 50 ng/mL for 16–18 h. Cells were then collected into RIPA buffer and IL-6 signaling, and Axl expression was analyzed by Western blotting. Expression levels of miR-34a were analyzed by RT-PCR as described in the previous section.

### 2.8. Statistical Analysis

Statistical analyses were performed with the Prism 9.2 (Graphpad, San Diego, CA, USA). Luciferase activity assays were analyzed with unpaired two-tailed student *t*-test with Welch’s correction. Western blot results were analyzed using Image Studio software. Intensity differences between groups were compared using the Mann–Whitney U test. Data are shown as median with interquartile range. A *p*-value < 0.05 was considered statistically significant.

## 3. Results

### 3.1. Multiple TFs Regulate Axl Expression in Mesangial Cells

Previously, we identified Sp1 as one TF that promotes Axl expression in renal mesangial cells [5]. In this study, candidate TF binding sites were identified using the PROMO and pDRAW32 software available online. Six TFs (Sp1, AP1, Ep300, C-Rel, STAT3, and MZF1) satisfying the maximum 15% dissimilarity were identified (Figure 1A). Further analysis revealed three types of secondary structures in a 600 bp region proximal to the *Axl* ATG translation start site (Figure 1B). These potential secondary structure-forming regions include a 30 AG microsatellite repeat, a mini-intron, and several candidate G-quadruplex, all of which are well studied structures known to regulate gene transcription. Interestingly, four of the six TFs have at least two predicted binding sites in the 600 bp proximal region (Figure 1B).

To investigate the importance of the TF binding sites, we performed two sets of experiments. We first treated cultured mesangial cells with siRNAs that target individual TFs and measured Axl expression by Western blot. As shown in Figure 1C, siRNA mediated knockdown of TF expression diminished Ep300, Sp1, and AP1. SiRNA targeted MZF1 knockdown was not complete. Knockdown of TFs AP1 and Sp1 also decreased MZF1 expression to the similar level as siRNA MZF1-targeted knockdown (Figure 1C). Nevertheless, four TFs, Ep300, AP1, Sp1, and MZF1, were verified to directly regulate Axl expression as decreased levels of Axl protein were detected by Western blot when mesangial cells were treated with the cognate TF-targeted siRNAs (Figure 1C). C-Rel and STAT3 were two TFs also identified with the same software. SiRNA treated mesangial cells exhibited a complete knockdown of the protein levels of C-Rel, but Western results showed no alteration of Axl expression (Figure 1D), suggesting that C-Rel, under the current culture conditions, does not regulate Axl expression in the mesangial cells. Similar results were obtained with STAT3 siRNA treatment (Figure 1D). Taken together, we identified AP1, Sp1, MZF1, and Ep300 as four main TFs that regulate Axl expression in renal mesangial cells.

We next PCR amplified the 4.4 kb of *Axl* 5′-UTR from the C57BL/6 mouse genomic DNA preparation and cloned this fragment into the pGL4.0 luciferase reporter vector. The full length of *Axl* promoter was sequence verified (Appendix A). Nine deletion constructs were generated subsequently with sequential truncation of 1 to 4.2 kb region from the original construct through restriction enzyme digestion (Figure 2A). The sizes of the individual deletion constructs were verified by restriction digestion (Figure 2B). To confirm our findings of the TF-regulated *Axl* expression, a full length (Axl-UTR) vector was co-transfected with individual TF vectors into Hela cells. Luciferase activities were recorded and analyzed over the baseline pGL4.0 vector. Luciferase activity increased ~5-fold in Hela cells co-transfected with mouse Sp1 compared to the luciferase activity transfected with the Axl-UTR vector alone (Figure 2C). This result is consistent with Sp1 regulated Axl expression in human cancer cells [8] and confirmed the Sp1 siRNA knockdown results in Figure 1C. Co-transfection with either mouse MZF1 or Ap1 also resulted in a significant increase of luciferase activity compared to the Axl-UTR alone (Figure 2C). Consistent with the siRNA inhibition data, mouse STAT3 co-transfection showed no difference in luciferase activity compared to the Axl-UTR transfection alone. Taken together, we have identified four TFs that contribute to Axl upregulation in mesangial cells with Sp1 being the most potent TF that upregulates Axl expression.

### 3.2. Proximal 600 bp Is Sufficient for Basal Axl Expression

To further delineate the minimally required core elements for full basal *Axl* expression, we transfected the nine serial-deletion luciferase constructs (Figure 2A) separately into Hela cells. All constructs, except the −0.2 kb, sustained similarly higher luciferase activity compared to the full length 4.4 kb *Axl*-UTR construct (Figure 2D). Five constructs (−3.2 kb, −1.7 kb, −1.4 kb, −1.1 kb, and −0.6 kb) showed a ~2-fold increased luciferase activity over the full-length promoter construct. Only one construct (−2.4 kb) showed similar luciferase activity over the full promoter construct, indicating additional regulatory element(s) presented in the −4.4/−3.2 kb region and/or the −3.2/−1.7 kb regions. The −0.2 kb promoter region showed a significantly decreased luciferase activity compared to the full length *Axl* promoter construct, suggesting that minimal promoter activity was supported by this 200 bp region (Figure 2D). This finding concurs with the information known about the four Axl transcription start sites (TSS1-4), which lie upstream of this region. Overall, the core *Axl* promoter activity lies within the GC-rich −600/−200 bp region predicted and proved by our data to bind to Sp1, AP1, and MZF-1 TFs (Figure 1 and Figure 2). Comparison of the Axl proximal promoter region (starting from the ATG start site of translation and ending ~800 bp upstream) between the C57BL/6 and BALB/C mouse genomes reveals few nucleotide differences in this region (BLASTn sequence alignment, data not shown). The identified TF binding sites, mini-intron sequence, and candidate G-quadruplex-forming regions in this Axl proximal promoter region are highly conserved between the two Mus musculus strains.

### 3.3. An Inhibitory Role of a Mini-Intron in the 5′-UTR of the Axl Promoter Region

In addition to TFs, an alternatively spliced mini-intron was discovered by transcription sequence alignment to lie within about 80 bp upstream of the *Axl* ATG translational start site (Figure 1B). Promoter introns can significantly increase gene expression levels through intron mediated enhancement [14]. We wondered if this mini-intron has a similar function in promoting *Axl* expression. Hela cells were transfected with the *Axl* promoter constructs with (Axl-UTR) or without the mini-intron (intronless, Axl-Δintron). Surprisingly, a ~3-fold increase in luciferase activity was recorded in Hela cells with Axl-Δintron compared to the luciferase activity in Hela cells with Axl-UTR (Figure 3A). We then repeated luciferase activity assays in Hela cells co-transfected with Axl-Δintron and various TF vectors. A similar pattern of elevated luciferase expression was observed compared to the Axl-UTR and TF co-transfection (Figure 3B compared to Figure 2C). Sp1 continued to be the strongest TF driving *Axl* transcription (Figure 2C and Figure 3B). MZF-1 slightly elevated luciferase activity though the difference between MZF-1 plus Axl-Δintron and Axl-Δintron alone was not significant (Figure 3B). Taken together, data presented here suggest an inhibitory role for the mini-intron in the *Axl* promoter region and its inhibitory function is independent of TF binding.

### 3.4. Transcriptional and Translational Inhibition of Axl by the G-Quadruplex Structure

We uncovered several previously unappreciated G-rich regions in the *Axl* promoter region (GR1-5, Figure 1B), which have the potential of forming G-quadruplexes at the DNA or RNA level. Promoter G-quadruplexes are key gene regulation elements and may serve as high-affinity binding hubs for the recruitment of transcription machinery to modulate gene expression [15]. G-quadruplexes in the 5′-UTR region can have an inhibitory or activating role in the modulation of gene expression [16,17]. To assess the function of *Axl* promoter G-quadruplexes in regulating Axl expression, we treated the Axl-UTR luciferase vector transfected Hela cells with the cationic porphyrin TMPyP4, which is able to bind and stabilize G-quadruplexes [18]. Treatment with TMPyP4 led to a significant dose-dependent decrease of luciferase activity (Figure 4). Similar results were obtained when Hela cells were transfected with the Axl-Δintron construct (Figure 4), demonstrating an inhibitory role for G-rich regions in regulating Axl expression in mice. A roughly 3-fold consistent increase in the intronless constructs compared to the Axl-UTR constructs (Figure 4, comparing the correspondent bars between Axl-UTR and Axl-Δintron) indicates that the upstream G-rich regions function independently of the mini-intron.

### 3.5. 3′-UTR Binding by miRNA Inhibits Axl Expression

MiR-34a has been shown to regulate Axl translation through direct binding to the 3′ UTR of the *Axl* mRNA transcript, and miR-34a expression is inversely correlated to the expression of Axl in breast cancer and colorectal cancer [19]. Here, we evaluated the role of miR-34a in the regulation of Axl in the kidney of nephritic mice and in cell cultures. Following induction of nephritis in B6 mice as previously reported [13], we examined miR-34a levels in total RNA samples extracted from the kidney cortex in nephritic mice eight days after nephritis development (Figure 5A). MiR-34a was found to be significantly downregulated in kidney samples from nephritic mice compared to kidney samples from WT non-diseased control mice (Figure 5B). This downregulated miR34a was associated with upregulated Axl expression in the kidney of nephritic mice, as shown in Figure 5C. Next, we treated mesangial cell cultures with increased concentration of miR-34a mimics. As shown in Figure 6A,B, Axl protein levels decreased with increased concentration of miR-34a in the culture, supporting an inhibitory role of miR-34a in Axl expression.

IL-6 is a cytokine well-known to promote pathologic renal inflammation [20]. IL-6 activation of STAT3 was found to downregulate miR-34a in cancer cells [21]. We wondered if increased levels of IL-6 in inflamed kidney stimulate Axl expression through the STAT3-targeted miR-34a inhibition. Primary kidney cells from B6 mice were purchased from the Cell Biologics (Chicago, IL, USA) and treated with rmIL-6. As shown in Figure 6C,D, rmIL-6 activated STAT3. STAT3 phosphorylation was associated significantly increased levels of Axl expression (Figure 6E). In the similarly rmIL-6 treated cells, significantly decreased miR-34a levels were also detected (Figure 6F). We, therefore, revealed a novel pathway involving the 3′-UTR of *Axl* that upregulates Axl expression in the kidney under the IL-6/STAT3 inflammatory conditions.

## 4. Discussion

Receptor tyrosine kinase Axl has been reported to be activated and contributed to proliferative renal diseases in kidney disorders in humans, rats, and mice [6,22,23,24]. However, Axl expression is not detected in the kidney of healthy individuals and naïve mice, but only upregulated under inflammatory conditions [25]. The primary cause for the increased expression of Axl in the kidney is not well-understood. As a granular understanding of the drivers of increased Axl expression might lead to basic insights into the primordial triggers for the underlying inflammation, we aimed in this study to identify the key regulatory domains with the Axl promoter region in mice. We have found that kidney Axl expression is complicated being regulated with a GC rich region in the 5′-UTR positively by TF initiated transcription and G-quadruplex mediated enhancement vs. negatively by mini-intron. Of particular interest and consistent with the literature on IL-6 and miR-34a, a key regulatory mechanism that may account for Axl-mediated renal inflammation is miR-34a binding to the 3′-UTR and suppression of Axl expression.

Axl expression in human cancer cells has been extensively studied and several TF binding sites were identified to regulate Axl expression in those cells [8]. Of all the TFs, Sp1 was found to be the most potent TF that drives Axl expression. We analyzed the mouse Axl promoter region with two different transcriptional analysis programs available online and found six candidate TFs (with maximum 15% dissimilarity) that may regulate Axl expression in mice. Out of these six TFs, four were verified by siRNA mediated knockdown and Western blotting. We planned to verify all four TFs with the luciferase reporter assay, but only three mouse TF expressing constructs were commercially available. Consistent with our previous finding [5] and published human cancer cell results [8], Sp1 is the strongest TF that drives Axl expression in mouse mesangial cells. Sp1 drives basal *Axl* transcription in solid cancer [8], but MZF-1 and AP-1 also drive Axl expression in leukemia cells [26]. All three TFs promote Axl expression in mouse mesangial cells. Additionally, we found that Ep300 is a strong TF that drives Axl expression in renal cells (Figure 1C). A minimum 600 bp upstream of the *Axl* transcriptional start site is sufficient to promote full transcriptional activity (Figure 2D). Surprisingly, a mini-intron (between −80 bp and −133 bp) was found to negatively regulate *Axl* transcription. There are at least four potential G-quadruplex secondary structures in the 600 bp proximal promoter region (Figure 1B). Stabilization of one or more of these structures with TMPyP4 significantly decreased luciferase activity indicating an impairment of Axl expression by candidate G-quadruplexes. Further work is needed to decipher the exact mechanism operating here.

The miR-34a binding site in the Axl 3′-UTR was first revealed in human cancer cells [19]. We found the same functional miR-34a binding site in the mouse Axl 3′-UTR. IL-6/STAT3-mediated miR-34a repression promotes and maintains invasiveness and metastasis in human colorectal cancer cells [21]. This effect may be due to the upregulation of Axl, which is highly expressed on cancer cells and promotes invasiveness and metastasis of many cancer cell lines [27]. Consistent with this paradigm, elevated IL-6 expression in the kidneys and urine of patients with mesangial proliferative glomerulonephritis is often associated with poor outcome [28]. IL-6 induces mesangial cell proliferation in the same study. Moreover, we found that Axl expression promoted mesangial cell proliferation in the mouse model of lupus nephritis [5]. Our results in Figure 6 showed for the first time that upregulation of Axl in primary kidney cells was stimulated by IL-6. Thus, we have revealed a novel pathway that IL-6 induction of STAT3 phosphorylation upregulates Axl expression in the inflamed kidney through inhibition of miR-34a.

The actual regulation of Axl expression in mice kidney is certainly more complex than we have revealed here. It is highly dependent on the inflammatory environment, a combination of inflammatory cytokines, i.e., IL-6 (Figure 6C) and other immune stimuli, i.e., IFN-γ [29]. In the Axl-UTR upstream of the Axl translational start site, there are total 332 TF binding sites identified by the software (data not shown). It is impossible to test all of these TFs due to the financial burden and lack of the siRNA and TF expressing constructs. Epigenetic modification of the Axl promoter is another possible mechanism to regulate Axl expression. Seventeen CCWGG methylation sites were identified in the mouse Axl promoter region by sequence analysis. The same number of methylation sites was reported in the human Axl promoter region [8]. CCWGG methylation has been shown to be associated with transcription silencing [30].

There are several limitations in the current study. ATCC mesangial cells are SV40 transformed immortalized stable cell lines, and they may not retain expression of transcription factors at the same levels as renal resident cells. It would be more relevant to test TF levels in primary mesangial and tubule cells, where Axl was found to be upregulated in vivo. However, primary renal cells are not broadly commercially available; they are difficult to culture (allow for two passages). Transfection of those cells with TF plasmids leads to low viability. Isolation of primary renal cells in the lab encountered with technical issues. Single cell TF and Axl expression analysis may provide with on-site association between Axl upregulation and TF expression. Results may help to find the perfect window for Axl-targeted intervention.

Taken together, this is the first study to characterize the mouse *Axl* gene promoter and to identify four essential TFs and G-rich key secondary structures as major elements of *Axl* gene expression and regulation in mouse kidney cells. Further study is needed to elucidate the function of these elements and other TFs in the inducible expression of Axl.

## Figures and Tables

**Figure 1 cells-11-01869-f001:**
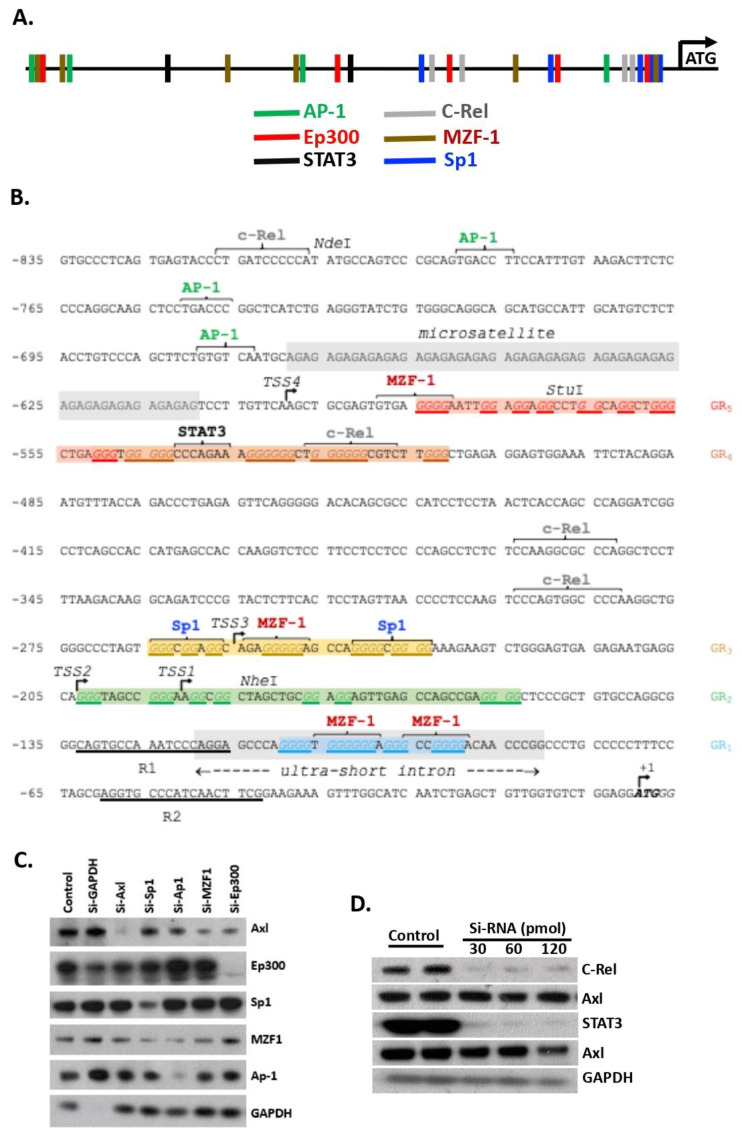
Axl expression is regulated by multiple transcription factors in renal mesangial cells. (**A**) characterized potential TF binding sites; (**B**) schematic representation of the 5′ flanking region of Axl proximal promoter with putative TF binding sites. Secondary structures are highlighted. TSS, transcription start site. GR, potential G-rich quadruplex regions; (**C**,**D**) SV40-mesangial cells were treated with siRNAs (Sp1, Ap-1, MZF1, and Ep300 in (**C**); C-Rel and STAT3 in (**D**)). Expression levels of Axl, and transcription factors were analyzed by Western blot. GAPDH was used as a loading control and internal siRNA control. Data shown were representative of three repeats.

**Figure 2 cells-11-01869-f002:**
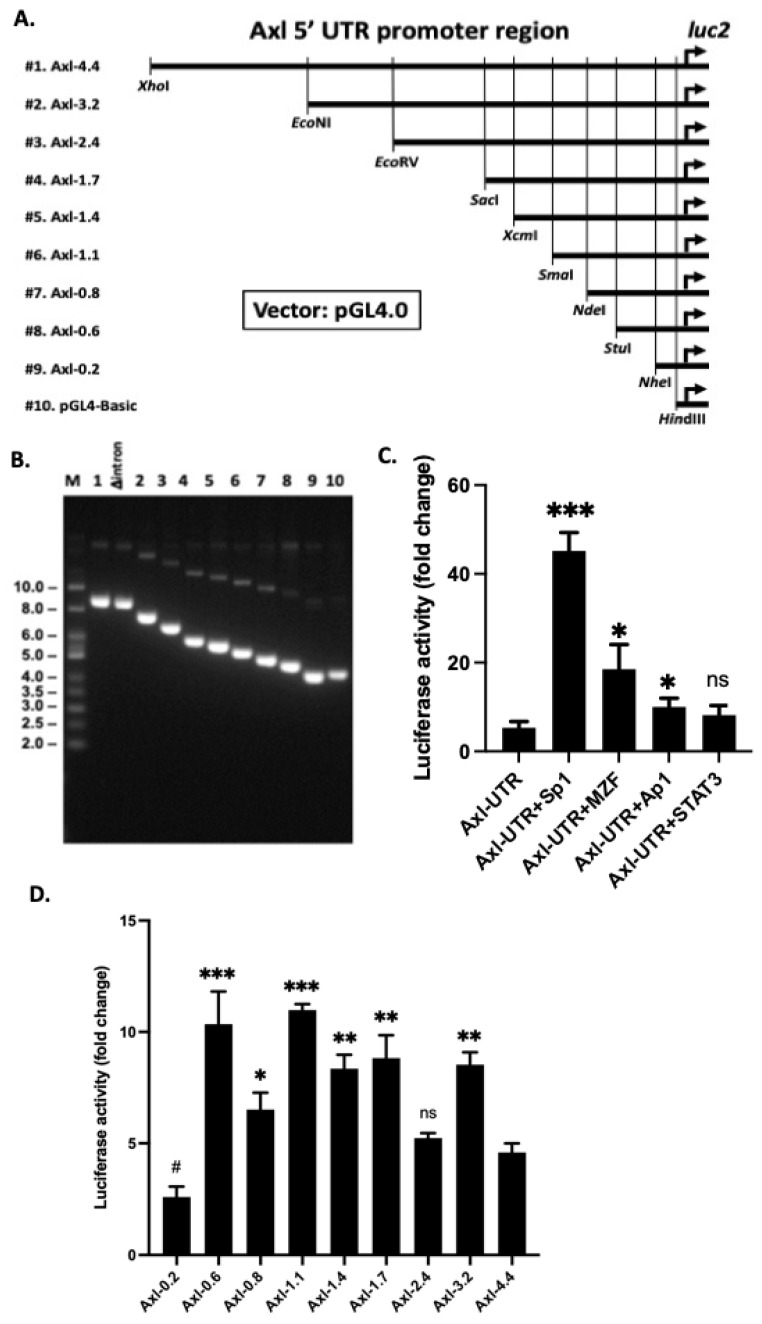
Axl promoter region constructs and luciferase assay. (**A**) schematic representation of serial deletion constructs of the Axl promoter region; (**B**) gel electrophoretic analysis of the Axl promoter serial deletion constructs. HeLa cells were transient transfected with various pGL4-Axl-luc2 reporters, with (**C**) or without (**D**) transcription factor expression vectors. Luciferase activity was assayed 24 h after DNA transfection. Experiments were repeated at least twice. Student t-test was performed to compare all constructs to the same full length Axl construct, Axl-UTR (**C**) or Axl-4.4 (Axl-UTR) (**D**). * *p* < 0.05, ** *p* < 0.01, *** *p* < 0.001, # *p* < 0.05, ns, not significant. The asterisk sign indicates significantly higher. The pound sign indicates significantly lower.

**Figure 3 cells-11-01869-f003:**
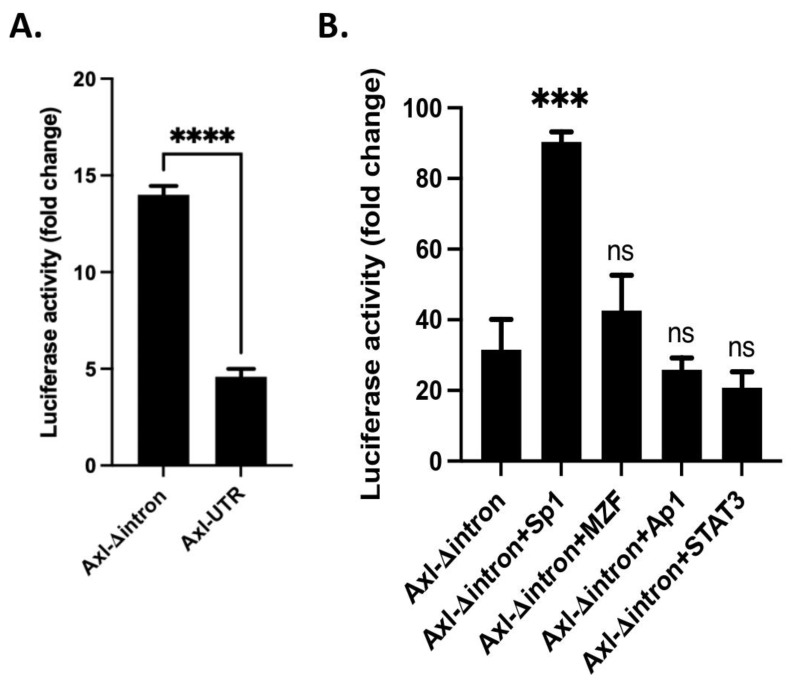
Inhibitory role of mini-intron in Axl promoter region. Hela cells were transfected with the Axl-Δintron constructs alone (**A**) or together with the transcription factor vectors (**B**) as described in the methods. Experiments were repeated three times. Luciferase activity was analyzed, and results were presented as mean ± SD. Student *t*-test was performed to compare with the luciferase activity from Hela cells transfected with the Axl-Δintron construct. *** *p* < 0.001, **** *p* < 0.0001, ns, not significant.

**Figure 4 cells-11-01869-f004:**
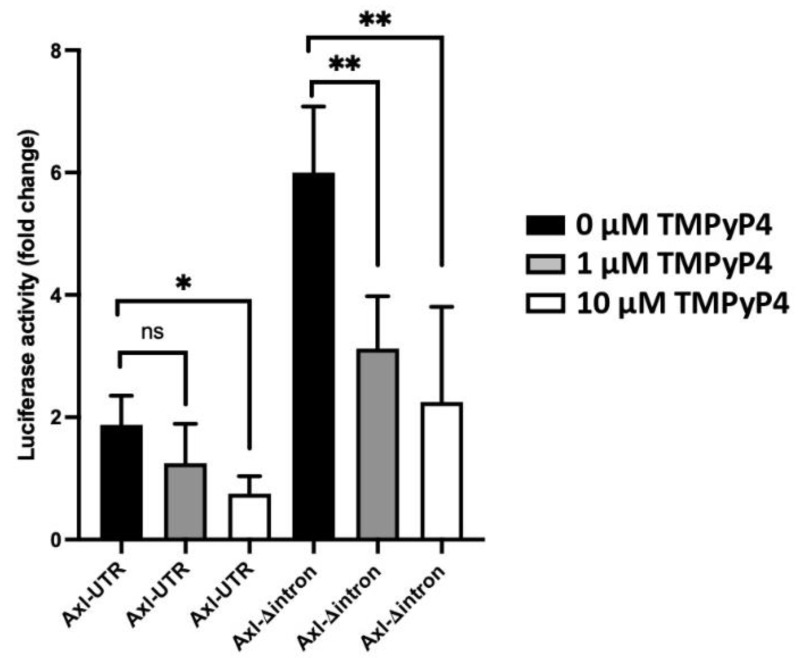
G-quadruplexes facilitates Axl expression. Hela cells were transfected with the Axl-UTR or Axl-Δintron constructs. The G-quadruplex stabilizer TMPyP4 was added into the culture two hours after transfection at different concentrations as indicated in the figure. Experiments were repeated twice. Luciferase activity was analyzed and results were presented as mean ± SD. Student *t*-test was performed to compare with the luciferase activity from Hela cells transfected with the control. * *p* < 0.05, ** *p* < 0.01, ns, not significant.

**Figure 5 cells-11-01869-f005:**
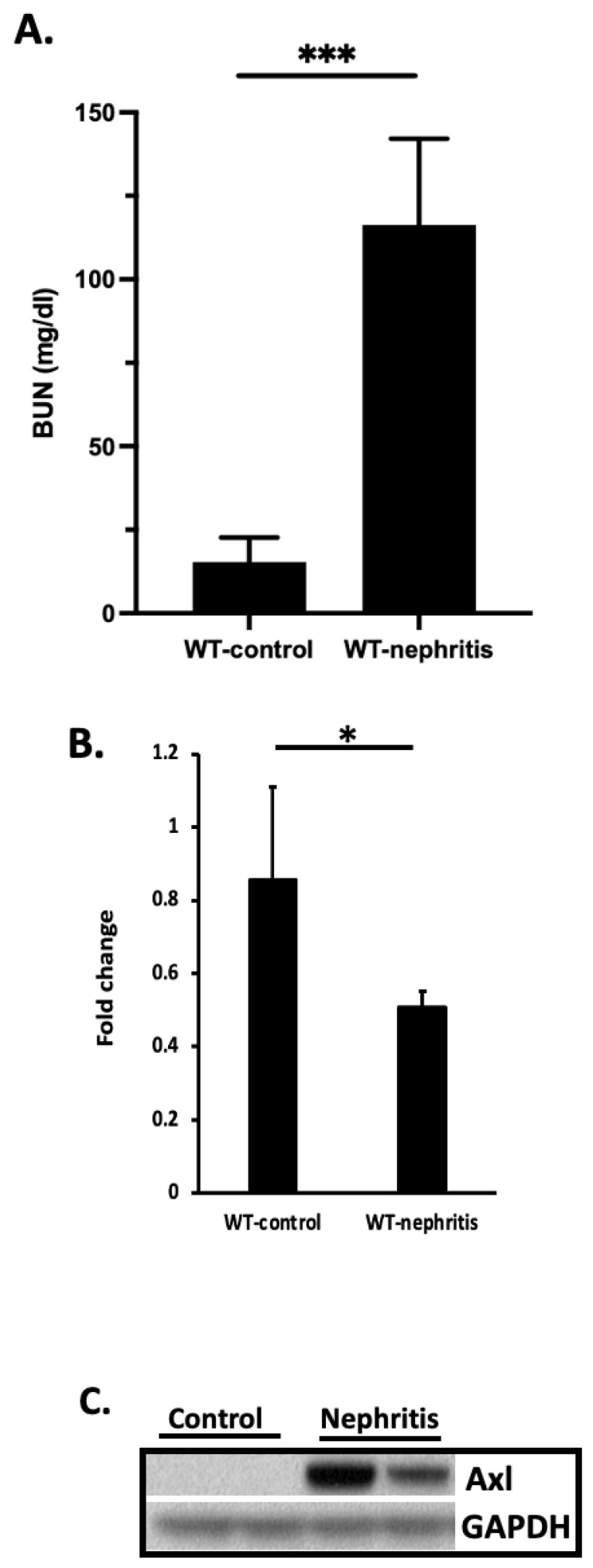
MiR34a levels inversely correlates with Axl expression. (**A**) Glomerulonephritis was induced. Kidney function was evaluated by measuring BUN levels. *** *p* < 0.001 by Student *t*-test. (**B**) Expression of miR34a in the kidney of experimental animals was analyzed by RT-PCR. * *p* < 0.05 by the Student *t*-test. Data are representative of two individual experiments. A total of seven mice were analyzed for miR34a expression. (**C**) Axl expression levels were analyzed by Western blot from the same kidney preparation.

**Figure 6 cells-11-01869-f006:**
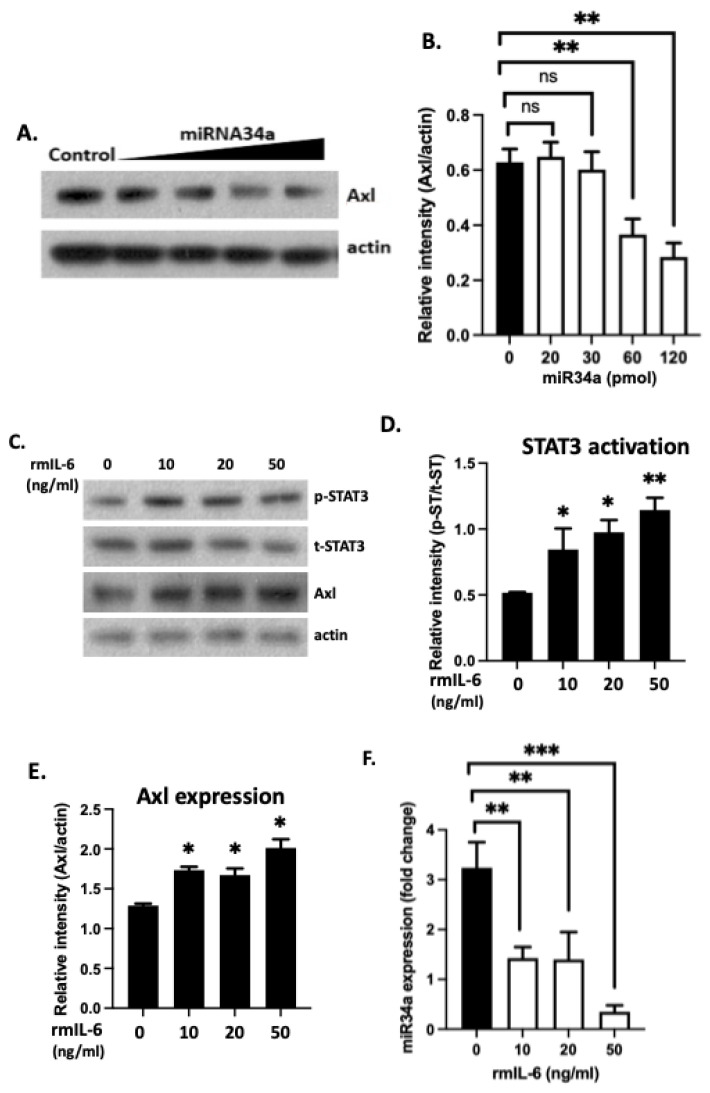
miRNA34a is a negative Axl regulator in mesangial cells. (**A**) Mesangial cells were treated with increased concentrations miRNA34a mimics. Cells were lysed in RIPA buffer after two days. Axl expression was analyzed by Western blot. Beta actin blot serves as an internal loading control. (**B**) Axl expression levels were quantified. (**C**) Primary kidney cells were treated with increased concentration of rmIL-6 for 16 hrs. Cells were then lysed in RIPA buffer and analyzed for STAT3 activation and Axl expression. Quantified data are presented in (**D**,**E**), respectively. (**F**) Expression of miR34a from the IL-6 treated primary kidney cells was analyzed. Experiments were repeated twice. Mann–Whitney U test. * *p* < 0.05, ** *p* < 0.01, *** *p* < 0.001, ns, not significant.

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
