# Peer review of "Axl Expression in Renal Mesangial Cells Is Regulated by Sp1, Ap1, MZF1, and Ep300, and the IL-6/miR-34a Pathway"

_cells, 2022, doi:10.3390/cells11121869_

Round 1
Reviewer 1 Report
The manuscript titled “Axl expression in renal mesangial cells is regulated by transcription factors and the IL6/miRNA34a pathway” is a collection of in vitro and in vivo studies interrogating the transcriptional regulation of Axl in inflammatory kidney disease.
Suggestions with potential to significantly increase the impact of the manuscript:
[1] Page 2, lines 70-75: Only one source of mouse mesangial cells were used for Figure 1. Authors should include the strain of mice used to derive the MC available from ATCC. It would be important to reproduce at least some of the findings in primary mesangial cells from at least one additional mouse strain.
[2] It is likely that genomic DNA sequence exists in genomic databases for several mouse strains. Are the TF binding sites, the mini-intron sequence, and the G-quadruplexes conserved across mouse strains? If sequence is available for murine lupus strains (MRL, NZB, NZW, etc.), it might be interesting to see what promotor polymorphisms exist.
[3] The putative G-quadruplexes shown in figure 1 are restricted to the upstream 600bp, but the luciferase assays used 4.4kb upstream sequences. Were there any additional putative G-quadruplexes distally between 600 and 4000bp upstream of Axl gene?
[4] Figure 5: There is no data provided to assess the successful induction of nephrotoxic serum nephritis in the mice used for data in this figure. This figure should only include the in vivo data from the nephrotoxic serum nephritis model, and data on proteinuria or histopathology (or at least immune complex deposition) should be added to help reader assess the degree of nephritis. If the degree of mesangial hypercellularity varied from mouse to mouse in their in vivo model, perhaps it correlates with the degree of Axl or miR-43a up- or down-regulation. The in vitro data on miRNA34a mimics and rmIL6 should be moved to a dedicated figure.
Minor issues / suggestions:
[5] Title: would “IL6/miR-34a” be more accurate than “IL6/miRNA34a?” It might also be better to include the names of the TFs in the title, rather than just stating that “Axl … is regulated by TFs and …”
[6] What is the difference between Axl-UTR and Axl-4.4 ? Which was used in figures 3A and 4?
[7] page 5, line 156: please rephrase this sentence which is awkward: “The level of MZF1 was at the modest“
[8] Figure 1C: either use densitometry for generating semi-quantitative results, or revise results section to report western blot results as qualitative (avoid phrases such as “dramatically decreased” and “almost completely diminished”)
[9] Where does Ep300 bind on the AXL promotor?
[10] Discussion should include a paragraph listing the limitations of these studies. It should include the fact that primary culture cells may not retain expression of transcription factors at the same levels, especially after passaging and sub-culture.
[11] Introduction should state how tissue-specific the expression levels are for the TAM family members. Discussion should state why Axl expression is low in mesangial cells despite the presence of all identified transcription factors. What makes mesangial cells different than other glomerular cells (podocytes and endothelial cells), and different than tubulointerstitial pericytes outside the glomeruli)? Why is hypercellularity in glomerulonephritis restricted to inflammatory cells and mesangial cells, and do TAM family members mediate this specificity?
Author Response
[1] Page 2, lines 70-75: Only one source of mouse mesangial cells were used for Figure 1. Authors should include the strain of mice used to derive the MC available from ATCC. It would be important to reproduce at least some of the findings in primary mesangial cells from at least one additional mouse strain.
We thank the reviewer’s suggestion. The ATCC mesangial cells were established from normal 4-week-old mice (C57BL/6J x SJL/J) (Proc. Natl. Acad. Sci. 1992). We now included this information in the text. Primary mesangial cells are only available from the CellBiologics. They are hard to maintain in culture (maximum two passages). We transfected those cells, but never collected enough cells after 2-3 days for any experiment. One vial can only be used for one experiment (IL-6 treatment in Figure 5). We tried several times to isolate primary renal cells from mice in the lab, but encountered with technical issues.
[2] It is likely that genomic DNA sequence exists in genomic databases for several mouse strains. Are the TF binding sites, the mini-intron sequence, and the G-quadruplexes conserved across mouse strains? If sequence is available for murine lupus strains (MRL, NZB, NZW, etc.), it might be interesting to see what promotor polymorphisms exist.
Comparison of the Axl proximal promoter region (starting from the ATG start site of translation and ending ~800 bp upstream) between the C57BL/6 and BALB/C mouse genomes reveals few nucleotide differences in this region (BLASTn sequence alignment). The identified TF binding sites, mini-intron sequence, and candidate G-quadruplex-forming regions in this Axl proximal promoter region are highly conserved between the two Mus musculus strains. Sequences for murine lupus strains are not available.
[3] The putative G-quadruplexes shown in figure 1 are restricted to the upstream 600bp, but the luciferase assays used 4.4kb upstream sequences. Were there any additional putative G-quadruplexes distally between 600 and 4000bp upstream of Axl gene?
There were no additional putative G-quadruplexes exist between 600 and 4000 bp upstream of Axl gene.
[4] Figure 5: There is no data provided to assess the successful induction of nephrotoxic serum nephritis in the mice used for data in this figure. This figure should only include the in vivo data from the nephrotoxic serum nephritis model, and data on proteinuria or histopathology (or at least immune complex deposition) should be added to help reader assess the degree of nephritis. If the degree of mesangial hypercellularity varied from mouse to mouse in their in vivo model, perhaps it correlates with the degree of Axl or miR-43a up- or down-regulation. The in vitro data on miRNA34a mimics and rmIL6 should be moved to a dedicated figure.
We now included proteinuria in Figure 5A. We are investigating the correlation of Axl expression with miR34a levels in the nephritis mice and in SLE patients with lupus nephritis. We have not been able to establish a significant association. This is probably due to the small sample sizes. The project is still going on. We moved the in vitro data on miR34a mimics and rmIL-6 to the separate figure 6.
Minor issues / suggestions:
[5] Title: would “IL6/miR-34a” be more accurate than “IL6/miRNA34a?” It might also be better to include the names of the TFs in the title, rather than just stating that “Axl … is regulated by TFs and …”
We thank the reviewer’s suggestion. We now changed the title to “Axl expression in renal mesangial cells is regulated by Sp1, Ap1, MZF1, and Ep300, and the IL-6/miR-34a pathway”
[6] What is the difference between Axl-UTR and Axl-4.4 ? Which was used in figures 3A and 4?
They are the same. We now changed all of them to Axl-UTR.
[7] page 5, line 156: please rephrase this sentence which is awkward: “The level of MZF1 was at the modest“
We deleted this part, and keep the sentence as “SiRNA targeted MZF1 knock down was not complete.”
[8] Figure 1C: either use densitometry for generating semi-quantitative results, or revise results section to report western blot results as qualitative (avoid phrases such as “dramatically decreased” and “almost completely diminished”)
We revised the results section on western blot as qualitative, and deleted the words “dramatically” and “almost completely”.
[9] Where does Ep300 bind on the AXL promotor?
The co-transcriptional activator Ep300 does not bind to DNA directly, but it is recruited to promoters by other factors that recognize nucleic acid sequence motifs (Zhou P et al. 2017). The PROMO program predicts that there are several TFs that bind to the Axl proximal promoter 5’ UTR region potentially, and which are known also to interact with Ep300 (potential binding sites were labeled in Fig. 1A).
[10] Discussion should include a paragraph listing the limitations of these studies. It should include the fact that primary culture cells may not retain expression of transcription factors at the same levels, especially after passaging and sub-culture.
We thanks the reviewer’s suggestion. We included a paragraph to discuss the limitations of the studies and stated the limitation of studying transcription factor levels in mesangial cell cultures.
[11] Introduction should state how tissue-specific the expression levels are for the TAM family members. Discussion should state why Axl expression is low in mesangial cells despite the presence of all identified transcription factors. What makes mesangial cells different than other glomerular cells (podocytes and endothelial cells), and different than tubulointerstitial pericytes outside the glomeruli)? Why is hypercellularity in glomerulonephritis restricted to inflammatory cells and mesangial cells, and do TAM family members mediate this specificity?
We have included in the introduction the expression patterns of TAM receptors. In the discussion section, we highlighted the potential mechanisms that upregulate Axl expression in mesangial cells.
Reviewer 2 Report
In the article entitled “Axl expression in renal mesangial cells is regulated by transcription factors and the IL6/miRNA34a pathway”, the authors sought to identify the key regulatory domains with the Axl promoter region in the mouse. They found that kidney Axl expression is positively regulated by a GC rich region in the 5’-UTR by TF initiated transcription, in particular, Sp1 in mesangial cells. The G-quadruplex stabilizer TMPyP4 mediated enhancement of Axl by decreasing luciferase activity of Axl-UTR and Axl-intron. Mechanistically, IL-6 activates STAT3 phosphorylation and was significantly associated with increased levels of Axl expression. To note, miRNA34a is a key regulatory mechanism that may account for the Axl-mediated renal inflammation by miRNA34a binding to the 3’-UTR and suppression of Axl expression. The study is novel and contributes to knowledge in the field. In vivo data would strengthen the manuscript.
Major comments
- The caption for the figure 5A mentions using at least 3 animals for the miRNA34a evaluation. The authors should consider increasing the sample size for a more consistent result.
- The authors showed in vitro the potential therapeutic effect of miR-34a. They should consider test it in vivo on their nephritis mouse model and show the inflammatory and histology alterations, as well as functional data (albuminuria and serum creatinine).
Minor comments
- The caption of figures 1D and E should include the concentrations of Si-C-Rel and Si-STAT3 used, as it is also not mentioned in materials and methods.
- Figure 5A: the caption describes the method for RNA extraction and evaluation. This should be kept in the materials and methods section.
- Figure 5D: reformulate its caption.
Author Response
- For figure 5A. The authors should consider increasing the sample size for a more consistent results.
We thank the reviewer’s comment. Data were actually “representative of 2 individual experiments”. At least 3 animals meant to say that total of 7 animals (set one, 3 animals; set two, 4 animals). We are sorry that we didn’t make it clear. We now stated that “Data are representative of 2 individual experiments. Total 7 mice were analyzed for miR34a expression”.
- They should consider test the therapeutic effect of miR34a in vivo on their nephritis mouse model and show the inflammatory and histology alterations, as well as functional data (albuminuria and serum creatinine).
We thank the reviewer’s suggestion. We are working on the therapeutic effect of Axl targeting. One direction is through miR34a. We are investigating a direct delivery of miR34a into the kidney to avoid potential adverse effects of systemic miR34a delivery. This should be beyond the scope of this manuscript.
- The caption of figures 1D and E should include the concentrations of si-C-Rel and si-STAT3 used, as it is also not mentioned in materials and methods.
We now included the concentrations in the figure 1D caption.
- Figure 5A. the caption describes the method for RNA extraction and evaluation. This should be kept in the materials and methods section.
We moved the method description part from figure 5A to the materials and methods.
- Figure 5D. reformulate its caption.
WE have reformulated the figure 5D caption to make it easier to understand.
Reviewer 3 Report
In the present study, Adams and co-workers were investigating transcriptional regulation of the receptor tyrosine kinase Axl in murine mesangial cells. They have characterized the effects of several transcription factors and secondary promoter structures on promoter activity. While the manuscript is well prepared and the presented results are conclusive, I have several technical/methodological concerns which should be addressed.
General
- Sequences for miRNA mimics . primers and siRNAs should be specified.
- An overview of the used antibodies and the used dilutions should be presented.
- The number of replicates and repeats should be given for the experiments.
- The used mesangial cell line should be specified. Passage numbers should be given.
- What kind of primary renal cells were used? Passage numbers should be given.
- Why were HeLa cells used for the luciferase assays? All other experiments were performed in murine cells and murine promotor was used. If the used mesangial cell line is not well transfectable, at least a murine cell line should be used. The passage of the used HeLa cells is missing.
Western blots
- Full blots should be shown as a supplement
- How much protein was loaded per lane?
- Quantification for Axl in the siRNA experiments is missing
- Why did the authors do increasing siRNA amounts for c-Rel and STAT3 (both had a more or less complete knockdown of the corresponding proteins), but not MZF1 where the knockdown efficiency was poor
- Why were different housekeeping proteins used in the experiments (Fig. 1 GAPDH and Fig. 5 actin)?
Cloning and luciferase assays
- The verified promoter sequence (as a supplement) or a link to the original sequence should be given.
- How was transfection efficiency quantified in the luciferase experiments? How were the results normalized? The size of the constructs is affecting the efficiency and there is a huge difference in the size of the deletion constructs.
- The authors present the results as a fold change in luciferase activity. What is used as the basal luciferase activity?
- In my opinion, Fig. 2C lacks sufficient controls. How is luciferase activity affected in cells co-transfected with pGL4.0 and the corresponding transcription factors?
miRNA expression
- Similar to the luciferase assays, the authors present the results in Fig. 5A and H as fold change. What was the control for the basal level of miRNA expression?
Do the authors have data for in vivo effects of miRNA34a antagomirs or mimics in the nephritis model?
Author Response
General
- Sequences for miRNA mimics. Primers and siRNAs should be specified.
Those are commercial products. We now included the catalog numbers in the methods part.
- An overview of the used antibodies and the used dilutions should be presented.
We thank the reviewer’s suggestion. We now included the antibodies and dilutions in the methods.
- The number of replicates and repeats should be given for the experiments.
We now included the replicates and repeats in the text and highlighted in yellow.
- The used mesangial cell line should be specified. Passage numbers should be given.
The mesangial cell line was from ATCC (please also refer to answer to Q1 from Reviewer #1). We now included the mesangial cell line and passage numbers (less than 5 passages).
- What kind of primary renal cells were used? Passage numbers should be given.
The primary renal cells are commercially available from the CellBiologics. The cells were directly used in the experiments. We don’t culture them in the lab.
- Why were HeLa cells used for the luciferase assays? All other experiments were performed in murine cells and murine promotor was used. If the used mesangial cell line is not well transfectable, at least a murine cell line should be used. The passage of the used HeLa cells is missing.
HeLa cells are a highly transfectable, human cell line commonly used for gene expression assays. They contain the basal mammalian transcription machinery that is needed for both mouse and human promoter studies. Because the human cells lack mouse TFs, it is possible to co-transfect these cells with Mus musculus TF expression vectors along with our mouse Axl-driven luciferase reporter to uncover species-specific TF effects. Use of mouse mesangial cells (without knockdown of endogenous TFs) would likely make it more difficult to see expression differences in the co-transfection studies (for example, endogenous TFs may overwhelm the reporter signal).
The laboratory’s HeLa cells were purchased from the ATCC. All experimental HeLa cell cultures were derived from a common stock. The HeLa cells were passaged every 3-4 days at 1-to-3 dilution. Each set of cells were passaged less than 10 times from the original freeze back and monitored biweekly for cell growth and health.
Western blots
- Full blots should be shown as a supplement.
Original full blots were uploaded with the manuscript.
- How much protein was loaded per lane?
We loaded 3mg of total protein.
- Quantification for Axl in the siRNA experiments is missing.
The expression of Axl by TF siRNAs is only indicative of possible involvement of these TFs. We further quantified their contribution of Axl expression in the luciferase assay. Quantitative analysis of Axl in the siRNA experiments are not necessary here.
- Why did the authors do increasing siRNA amounts for c-Rel and STAT3 (both had a more or less complete knockdown of the corresponding proteins), but not MZF1 where the knockdown efficiency was poor.
MZF1 siRNA efficiency is low compared to other siRNAs. We tried higher concentrations, but didn’t get improved knockdown. Other TF siRNAs (Ap1, Sp1 and Ep300) were introduced with the same concentration as MZF1. Those TFs were pulled together on one gel to better demonstrate Axl expressing levels attributed by those TFs. C-Rel and STAT3 (Fig.1D) were negative examples, therefore separated from the first set in Fig. 1C.
- Why were different housekeeping proteins used in the experiments (Fig. 1 GAPDH and Fig. 5 actin)?
We thank the reviewer for pointing out the discrepancy of internal controls. For most experiments, both GAPDH and actin were analyzed and similar results were obtained. In some cases, actin Ab was highly efficient. It’s hard to obtain a good exposure even with 1:1000000 dilution. Nevertheless, both GAPDH and actin were included as internal controls by numerous publications.
Cloning and luciferase assays
- The verified promoter sequence (as a supplement) or a link to the original sequence should be given.
We now included the original sequence of Axl promoter in supplement figure S1.
- How was transfection efficiency quantified in the luciferase experiments? How were the results normalized? The size of the constructs is affecting the efficiency and there is a huge difference in the size of the deletion constructs.
The Lipofectamine 3000-based transfection protocol recommended by the manufacturer was followed routinely (including amounts of added DNA relative to the total number of cells plated). Each transfection experiment was repeated 3-5 times. Expression results were normalized to the pGL4-Basic (empty vector) control in parallel transfection experiments. The amount of transfected DNA (i.e., Axl promoter-containing sequences) for the different promoter constructs (variable in size) was adjusted so that equal numbers of plasmid molecules were transfected per well with proportionally more amounts of the larger constructs added relative to the smaller constructs. All transfection and luciferase assays were performed on the same day using the same dilution of HeLa cells, employing the same master stock of luciferase reagent. The luminometer reads for individual plates were performed within 5 minutes of one another. Four separate luminometer reads were recorded per well to control for variability in the luminometer readings.
- The authors present the results as a fold change in luciferase activity. What is used as the basal luciferase activity?
Expression results were normalized to the pGL4-Basic (empty vector) control in parallel transfection experiments.
- In my opinion, Fig. 2C lacks sufficient controls. How is luciferase activity affected in cells co-transfected with pGL4.0 and the corresponding transcription factors?
We did the luciferase assay with TFs co-transfected with pGL4.0, and didn’t see alterations of the luciferase readout compared to the pGL4.0 alone. We also did the co-transfection with the empty vector (vectors without TF coding sequence), and did observe any difference between the Axl-UTR with or without TF empty vector.
miRNA expression
Similar to the luciferase assays, the authors present the results in Fig. 5A and H as fold change. What was the control for the basal level of miRNA expression?
MicroRNA expression was analyzed differently from the luciferase assay. The results were normalized using the 2-ΔΔCt method relative to the mouse kidney specific miRNA normalizer (481757_mir) recommended by the ThermoFisher technique support. The CT of each test message was first normalized using the CT for 481757_mir, assayed in the same sample. Fold changes were then calculated using the relative CT method: fold change = 2(normalized CT from experimental mice – normalized CT from control mice).
Do the authors have data for in vivo effects of miRNA34a antagomirs or mimics in the nephritis model?
As responded to reviewer #1’s suggestion, we are working on the therapeutic effect of Axl targeting. One direction is through miR34a. We are investigating a direct delivery of miR34a into the kidney to avoid potential adverse effects of systemic miR34a delivery. This is beyond the scope of this manuscript.
Round 2
Reviewer 1 Report
This reviewer would like to thank the authors for their responsiveness to the initial round of peer review. Some of the answers to peer review should be included in the manuscript, such as "Comparison of the Axl proximal promoter region (starting from the ATG start site of translation and ending ~800 bp upstream) between the C57BL/6 and BALB/C mouse genomes..."
No further suggestions.
Author Response
We thank the reviewer's suggestion. We now included this statement in the text.
Reviewer 2 Report
1. Yet the therapeutic effect of mir-34 is not available, the authors should demonstrate the functional (albuminuria & creatinine) and structural (morphology) changes in relation to the nephritis model.
Author Response
We appreciate reviewer's comment and valuable suggestion. We are working on the therapeutics in spontaneous and inducible lupus nephritis through Axl and Axl-downstream targeting. One of the Axl selective small molecule inhibitors, Bemcentinib (also known as R428), seems to be a promising candidate. We showed preventative effects of this drug (J Autoimmun, 2018). We just submitted another manuscript demonstrating the therapeutic effects in mouse nephritis after disease onset. All these studies showed functional and structural improvement. Bemcentinib is the first Axl inhibitor approved by FDA in clinical trials for cancer patients. Our data may facilitate a quick application of this drug in lupus nephritis patients.
Reviewer 3 Report
The authors have responded to all my questions and remarks. However, I could not find the uploaded full blots. It might be helpful to include these also in the supplements.
The section regarding the used antibodies includes to many 'the'. This should be checked. In Figure 5 A, 'nephriitis' should be 'nephritis'.
I have no further questions.
Author Response
We thank the reviewer's instructional suggestions. We sent the original full blots in a zip file to the managing editor. We carefully checked the antibody section and revised accordingly. We also thank the reviewer for picking up the typo in figure 5A. We now corrected this in the text.